

# CopterSonde-SWX: Development of a UAS-based Vertical Atmospheric Profiler for Severe Weather

Antonio R. Segales[a], Tyler M. Bell[a], Abdullah Al Tasim[a,c], Aaron Quiroz[a,d], Jeremy Simms[a,d], Joshua Gebauer[a], and Elizabeth N. Smith[b]

[a]Cooperative Institute for Severe and High-Impact Weather Research and Operations, The University of Oklahoma, Norman, Oklahoma
[b]NOAA/OAR National Severe Storm Laboratory, The University of Oklahoma, Norman, Oklahoma
[c]School of Engineering, The University of Oklahoma, Norman, Oklahoma
[d]School of Meteorology, The University of Oklahoma, Norman, Oklahoma

**Correspondence:** Antonio R. Segales (tony.segales@ou.edu)

**Abstract.** Growing demand for high spatiotemporal resolution observations in the planetary boundary layer (PBL) has driven the development of affordable, small uncrewed aircraft systems (UAS) technology to fill critical observational gaps and support improved understanding and future assimilation into prediction models. This work presents the CopterSonde-SWX (CSWX), an in-situ UAS vertical profiler that combines a high-thrust tilted-body airframe, a wind-vane flight mode for sampling undisturbed air, and a shielded actively ventilated sensor scoop to acquire thermodynamic and full 3D wind vectors without dedicated anemometers. Through a series of flow simulations and intercomparison field campaigns, including colocated flights with its predecessor (CS3D), Doppler wind lidars, and radiosondes, the CSWX demonstrated inter-sensor temperature uniformity within $\pm0.2°$C across variable solar and wind regimes and achieved LESO-based wind retrieval RMSEs of 0.49 m s$^{-1}$ (vertical) and 1.03 m s$^{-1}$ (horizontal). In a nocturnal low-level jet case, the CSWX sustained winds up to $\sim 24$ m s$^{-1}$ at 520 m (versus the CS3D's 20 m s$^{-1}$ limit at $\sim 275$ m), with polynomial fits projecting a safe maximum wind tolerance of 29.5 m s$^{-1}$ while retaining sufficient battery energy margin for safe return. These results support the CSWX as a resilient, high-fidelity platform for atmospheric profiling, advancing the transition from a research prototype to an operational instrument. Even though the CopterSonde is an experimental design, this work may serve as a guideline to define future standards for WxUAS development.

## 1 Introduction

Accurate, high-resolution observations of the planetary boundary layer (PBL) are essential for improving weather forecasting and climate modeling. Conventional measurement methods—such as ground-based sensors and radiosondes—have long provided valuable atmospheric data but are limited by infrequent sampling and coarse spatial resolution (Geerts et al., 2018; Dabberdt et al., 2005). In particular, radiosondes, which are typically launched only a couple of times a day at low spatial density, cannot resolve the rapid, localized changes that occur within the PBL. This observational gap poses a significant challenge for capturing the fast mesoscale evolution that is often key to predicting severe weather phenomena. A better understanding of



the atmospheric processes could be beneficial to society, given the profound socio-economic impacts associated with severe weather events. This includes thunderstorms, tornadoes, and tropical storms that cost billions of dollars annually to the U.S. economy (Lazo et al., 2011; Goklany, 2009; Webster, 2013). Changes in severe weather patterns further amplify these impacts

by intensifying damaging weather extremes (Liu, 2017; Frame et al., 2020), affecting biodiversity (Scheffers and Pecl, 2019), food production (Rosenzweig et al., 2001; Gregory et al., 2005), supply chains (Markolf et al., 2019), and public health (Curtis et al., 2017; Mirza, 2003).

Recognizing these challenges, meteorological organizations have underscored the need for innovative observation technologies. Initiatives by the National Weather Service (NWS) and the National Severe Storm Laboratory (NSSL) advocate for

integrated, high-resolution sensor networks to better monitor the lower atmosphere (NWS, 2019; National Severe Storm Laboratory (NSSL), 2015; Zhang et al., 2016). Recent advances in uncrewed aircraft systems (UAS) have opened promising avenues for atmospheric sampling. Fiebrich et al. (2021) further shows that a network of weather-sensing UAS (WxUAS) profilers could potentially deliver high-resolution vertical atmospheric data to improve forecasting, optimize agriculture and energy use, manage wildfires and air quality, and support climate monitoring, thereby yielding significant economic, environmental, and public

health benefits. Driven by progress in automation, affordable construction materials, and innovative design techniques, UAS technology has found success in fields such as agriculture, land surveying, and surveillance (Shakhatreh et al., 2019). Yet, while the commercial market has predominantly focused on these applications, the potential for UAS in meteorological research and operations, especially for high-frequency vertical profiles, has remained largely underexploited.

In response to this pressing need, this article introduces the next generation of CopterSonde UAS. The CopterSonde is a novel

WxUAS concept that is constantly evolving for in situ thermodynamic and kinematic profiling of the PBL. The CopterSonde diverges from conventional UAS designs by integrating a dedicated thermodynamic sensor package with an aspiration system and by incorporating custom flight control algorithms to mitigate platform-induced measurement errors (Segales et al., 2020; Bell et al., 2020; Greene et al., 2018, 2019). Furthermore, its onboard weathervane-like function enables the collection of high-fidelity thermodynamic and kinematic data without reliance on bulky and costly wind sensors (Segales, 2022). This

design concept preserves the CopterSonde's streamlined aerodynamic profile and minimizes aerodynamic drag. Altogether, these features make the CopterSonde uniquely suited to capture detailed vertical profiles of the PBL, particularly in high-wind conditions, thereby mitigating some of the limitations of traditional in situ techniques.

Collaborative field experiments across multiple institutions compared various WxUAS platforms alongside conventional meteorological instruments (de Boer et al., 2024; Hervo et al., 2023; Barbieri et al., 2019; Jacob et al., 2018; Koch et al.,

2018; Kral et al., 2018). These experiments have guided the evolution of WxUAS designs and shown that UAS-based sampling effectively bridges observational gaps by delivering flexible, high-frequency, and targeted atmospheric profiles. Many creative and effective WxUAS-based measurement techniques were tested and evaluated, with their results shared with the research community (Kistner et al., 2024; Wildmann and Wetz, 2022; Segales et al., 2022; de Boer et al., 2022; Islam et al., 2019; Houston and Keeler, 2018). Even the World Meteorological Organization (WMO) launched a global demonstration campaign

in 2024 to evaluate the capacity of emerging UAS technologies to satisfy future operational requirements (WMO, 2021). Datasets from some field experiments are available for analysis and verification (Bell and Segales, 2024; Greene et al., 2020;



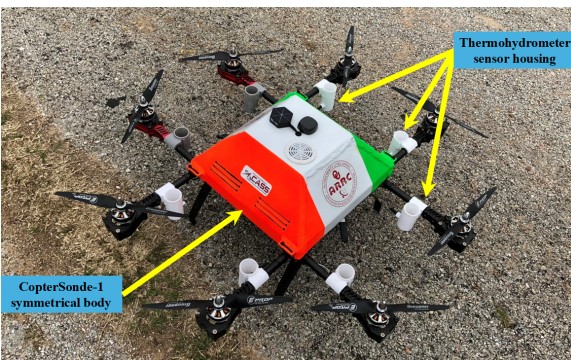

**Figure 1.** CopterSonde-1 concept made in early 2017. An H-frame octocopter UAS with thermohygrometer sensors distributed in a symmetrical arrangement around the main body. The atmospheric sensors were housed and protected inside white plastic tubes and mounted on the UAS's arms below the rotors, shown by the yellow arrows pointing to some of the sensors.

de Boer et al., 2020b; Lappin et al., 2023). The CopterSonde also contributed to the PERiLS linear-storm field campaign (Kosiba et al., 2024), yielding valuable datasets (Bell et al., 2024; Smith et al., 2024). During PERiLS, extreme ($> 25$ m s$^{-1}$) pre-storm winds exposed the platform's operational limits, showing how such conditions can hinder critical atmospheric data
collection for forecasting. We have thus determined that WxUAS require enhanced performance to operate in high-wind layers and reach deeper atmospheric profiles.

The work presented in this article is built upon the hypothesis that the CopterSonde system, a WxUAS-based in-situ vertical profiler tailored specifically for atmospheric sampling, can deliver measurements of comparable quality to conventional methods, while also providing enhanced temporal and spatial resolution and filling the observational gap in the PBL, even
in extreme wind conditions. The design and implementation of the CopterSonde draw on lessons from past field campaigns and extensive collaborative studies, including early prototype concepts and other further innovations in UAS meteorological applications (Wildmann et al., 2014; Chilson et al., 2019; de Boer et al., 2020a; McFarquhar et al., 2020). In doing so, the CopterSonde can not only help to fill the existing observational data gap but also pave the way for more adaptive and comprehensive weather sampling and monitoring systems, in line with strategic objectives from NOAA (NOAA, 2020) and the
recommendations outlined in recent decadal surveys (of Sciences, Engineering, and Medicine, 2018; Council, 2009).

The following sections detail the technological innovations, design considerations, and field validation experiments that underpin the development of the CopterSonde. This research leverages UAS capabilities to develop a reusable platform for reliable atmospheric data acquisition, potentially enabling its integration into numerical weather model data assimilation to improve forecast accuracy and deepen our understanding of PBL dynamics.
The remainder of this article is structured into five sections as follows. Section 2 traces the evolution of the CopterSonde from early prototypes to modular UAS concepts, detailing design drivers and prototype learnings. Section 3 describes the CopterSonde-SWX severe-weather edition, including its high-thrust propulsion, 3D-wind estimation, and thermodynamic sensor enclosure aimed at mitigating solar radiation effects. Section 4 summarizes field operations and observations, covering





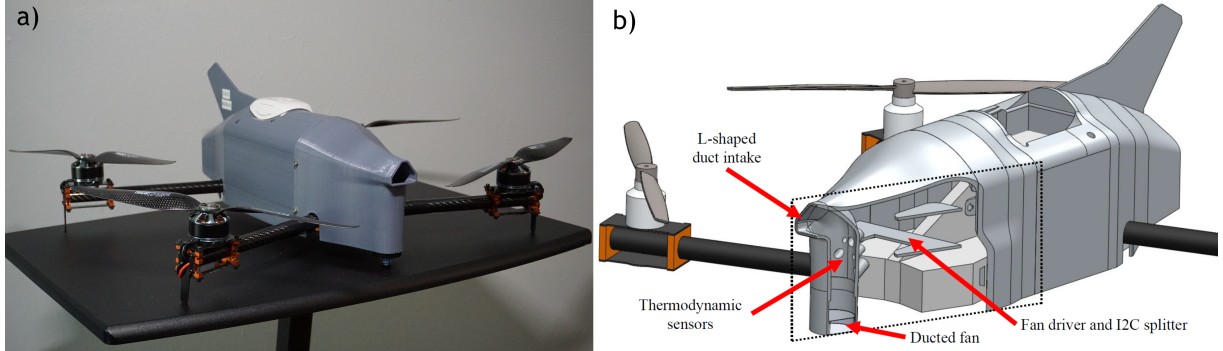

**Figure 2.** a) CopterSonde-3D concept developed in early 2022. This quadcopter UAS is designed for thermodynamic observations and vertical profiling of the lower atmosphere. Its distinctive shape results from an optimized internal layout that enhances airflow across the sensors via a wind vane flight mode. b) Section cut of the front shell from the CopterSonde-3D CAD model, illustrating the solar shield and sensor compartment design, along with the placement of the thermodynamic sensors and other critical components. The dotted rectangle marks the plane of the cut.

deployment procedures and validation experiments under high wind and in precipitation. Finally, Section 5 concludes by syn-
thesizing key findings on the CopterSonde-SWX (CSWX) design effectiveness, operational improvements, and demonstrated resilience in severe-weather environments, and outlines future development directions.

## 2  CopterSonde UAS Conceptualization

The evolution of the CopterSonde began as part of the ambitious 3D Mesonet project led by the University of Oklahoma between 2016 and 2021 (Chilson et al., 2019). The project aimed to enhance thermodynamic and kinematic measurements of
the lower troposphere by augmenting traditional 2D, tower-based observations with high-resolution vertical profiles obtained from UAS. Early prototypes, such as the CopterSonde-1 (CS1), see Fig. 1, adopted a standard multicopter configuration with a symmetrical arrangement of temperature and humidity sensors mounted below the rotors. Although these initial designs facilitated quick manufacturing and deployment, preliminary field tests revealed that placing atmospheric sensors in the rotor wash region of the UAS often compromised measurement quality (Greene et al., 2018). Overall, the underperformance of the
early CS1 design was noticeable in comparison to other platforms during field campaigns (Koch et al., 2018).

To address these issues, extensive experiments were conducted in controlled settings, where various onboard sensor locations were evaluated to minimize aerodynamic disturbances caused by the UAS. (Jacob et al., 2018; Greene et al., 2018, 2019). The results indicated that no single position in proximity to propellers could entirely eliminate contamination, thereby prompting a redesign focused on airflow management. Consequently, we opted to integrate a dedicated ventilation and solar shielding
system to maximize undisturbed air aspiration across the thermohygrometer sensors. We refer to this integration as the Copter-Sonde's scoop, which constitutes a centralized and modular sensor payload as depicted in Fig. 2b. The scoop and payload





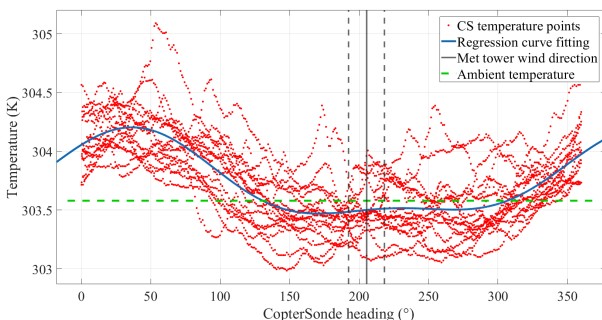

**Figure 3.** Scatter plot of temperature observations against the aircraft's heading collected with the CS3D while hovering and rotating with a constant angular speed of 2 revolutions per minute. The mean ambient temperature and wind direction were computed using observations from a nearby meteorological tower during the flight. The vertical dashed lines represent 3 times the standard deviation of the measured wind direction. The blue line represents the best-fit polynomial curve.

compartment were then molded and shaped to seamlessly blend with the external shell of the CopterSonde with the intent of improving aerodynamic performance. CAD software and 3D-printing techniques were used for fast development and pro-totyping of the CopterSonde's shell and scoop. A comprehensive description of these features is provided by Segales et al.
(2020) and Chilson et al. (2021). These features form a core aspect of the CopterSonde's design and development, extending to subsequent generations.

The adoption of the scoop would not have been feasible without the implementation of a custom wind vane flight mode (WVFM). The WVFM is a function implemented on the autopilot system to orient the CopterSonde into the wind, positioning the sensor compartment in the most upwind section of the aircraft to sample undisturbed air and guarantee the sensors operate
within their nominal conditions. Such iterative design refinements led to the development of the CopterSonde-3D (CS3D) UAS (shown in Fig. 2a). This comprehensive evolution—from the provisional CS1 to the advanced CS3D—exemplifies a robust engineering workflow that balances rapid prototyping with thorough testing in real-world scenarios.

Field experiments were conducted to demonstrate the CS3D design's functionality. These were specifically designed to validate sensor placement and to assess the effectiveness of the WVFM in mitigating undesired platform-induced heat sources
under real-world conditions. Windy days were chosen to increase the prominence of the heat advection effects produced by the CopterSonde's body, electronics, and motors. Flights were scheduled around solar noon to minimize the temperature bias caused by solar radiation on the sides of the platform. Stationary flight or hovering configurations were performed at the same height as the temperature probes from a nearby meteorological tower used for reference.

For this experiment, the CS3D was programmed to rotate continuously about its vertical axis at a constant angular speed
of 2 revolutions per minute. Through this configuration, the localized sensor arrangement of the CS3D effectively captured temperature measurements across all azimuthal directions. Figure 3 shows a point cloud of temperature observations plotted against the CS3D's heading. The best fit curve, derived through Fourier curve fitting, demonstrated a distinct wave-like pattern. Notably, the region corresponding to the minimum temperature—closely matching the mean ambient temperature obtained





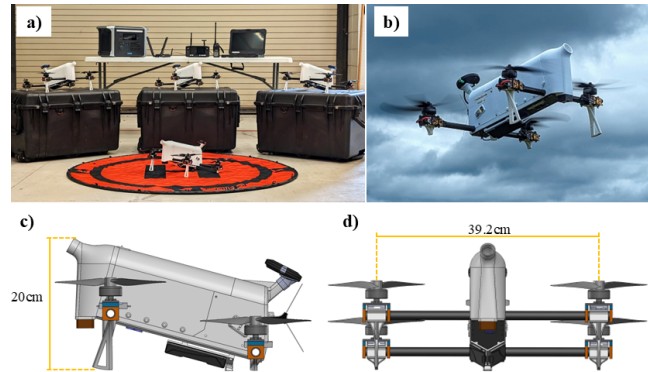

**Figure 4.** CopterSonde-SWX (CSWX) concept developed in 2024. Building on the success of the CopterSonde-3D, the CSWX is a weather-sensing UAS tailored for severe weather measurements and fabricated with a high-performance propulsion configuration. Panel a) shows a fleet of CSWX UAS sitting alongside its ground support equipment and transport cases. Panel b) captures an airborne CSWX on a stormy day. The bottom row presents the CSWX's side view (panel c)) and front view (panel d)), each annotated with key dimensions for scale.

from a nearby meteorological tower—was flattened over a range of approximately $50°$ on either side of the mean wind direction. This not only confirmed the high effectiveness of the sensor package design but also provided quantitative operational constraints for the WVFM. Based on these observations, a yaw tolerance range of $50°$ on either side was established as a design constraint for the wind vane mode, ensuring that the thermodynamic measurements remain reliable even with slight deviations in orientation.

These field observations have been critical in refining the CopterSonde design. They provide clear justification for the sensor placement strategy and the integration of the WVFM—key elements that distinguish later CopterSonde prototypes from the earlier designs and contribute to more reliable atmospheric data collection.

**Table 1.** Measured technical specifications that define the flight envelope's *maximum limits* of the CopterSonde-SWX obtained from a compilation of more than 65 flight tests in the highest-performance configuration. The values in parentheses are the theoretical maximums for the corresponding parameter.

| Specification | Value | Specification | Value | Specification | Value |
|---|---|---|---|---|---|
| All-up weight | 2.85 kg | Descent rate | 7 (10) m s$^{-1}$ | Min. air density | 0.95 (0.85) kgm$^{-3}$ |
| Hover time (10% batt.) | 15.6 min | Input voltage | 26 V | Altitude above ground | $1,500 \, (3,000)$ m |
| Forward top speed | 35.4 m s$^{-1}$ | Peak current | 110 (264) A | Temperature range | $-20$ to $45\,°$C |
| Mean wind tolerance | 25 (31.8) m s$^{-1}$ | Cont. current | 65 (140) A | Rel. humidity range | 0 to 100 % |
| Gust tolerance | 28 (35.4) m s$^{-1}$ | Motor Power | 610 (2100) W | Payload weight range | 300 to 450 gr |
| Climb rate | 5 (12.5) m s$^{-1}$ | Motor temperature | 65 (105) $°$C | Mechanical vibration | 30 ms$^{-2}$ |



## 3 CopterSonde-SWX — Severe Weather Edition

High-impact weather events often have extreme atmospheric conditions that many conventional UAS struggle to overcome.
Therefore, the impetus for the newest generation of WxUAS-based vertical profilers stems from our belief that these platforms
must outperform conventional UAS to measure extreme atmospheric conditions. Extensive field experience in these types of
environments shows that current systems often hit their performance limits—whether due to altitude range, wind tolerance,
temperature extremes, or energy consumption— and are thus unable to collect valuable data in harsh weather conditions.
In a networked system of drones, similar to the 3D Mesonet concept presented by Chilson et al. (2019), understanding these
platform limits is crucial for reliably maximizing high-fidelity data collection for research and weather modeling. Consequently,
in addressing these challenges, we have prioritized the development of a high-power platform that can endure UAS-hostile
conditions while incorporating robust safety diagnostics and weather-dependent operational modes.

Building on the success of the CS3D (Segales et al., 2020) and targeting severe weather measurement and high-resolution
vertical profiling of the PBL, the CopterSonde-SWX (CSWX), shown in Fig. 4, was engineered with a high-performance
propulsion system and aerodynamic configuration. Under this design philosophy, the CSWX sacrifices some flight endurance
in favor of superior power and operational robustness. This trade-off enables the platform to push its performance to the
limits—capturing dynamic atmospheric phenomena where traditional radiosondes and lower-performance UAS platforms may
fall short. However, we are confident that future advancements in high-capacity, lightweight battery technology (Hasan et al.,
2025; Itani and De Bernardinis, 2023) will eventually restore and even enhance flight endurance without sacrificing these
high-power specifications. Ultimately, the goal of the CSWX is to mark a significant step forward in developing resilient,
high-impact WxUAS for advanced atmospheric research and, subsequently, future operational networks.

### 3.1 Airframe and Propulsion System

Similar to its predecessor, the CSWX is a quadcopter featuring an in-house-manufactured airframe built from carbon-fiber
plates connected by aluminum standoffs. The airframe is elongated, narrowed, and angled upward to achieve high aerodynamic
performance and reduce air resistance under strong wind conditions. Figure 5 illustrates the updated SWX airframe both in
a stationary hover configuration (or resting on the ground) and pitched forward during forward flight or to counteract strong
winds. The figure also depicts the relative positions of external components, such as the GNSS and air intake, aligned with the
wind direction typically experienced during vertical atmospheric profiling.

The aircraft's outer shell was 3D-printed using polycarbonate plastic, a material substantially stronger than the previously
used PLA. Polycarbonate also provides superior resistance to debris impact and can withstand ambient temperatures up to
approximately $80°$C before warping. The arms consist of single-piece, hollow carbon-fiber tubes, through which wiring is
internally routed to the motors. These tubes allow for rotational adjustment of the rotor orientation, enabling changes in the
inclination of the aircraft's main body relative to the propeller plane.

At the end of each arm is a 3110-size 900KV brushless motor, driven by a high-performance electronic speed controller
(ESC) capable of handling a maximum continuous current of 80 A. To ensure effective cooling, the ESCs were positioned





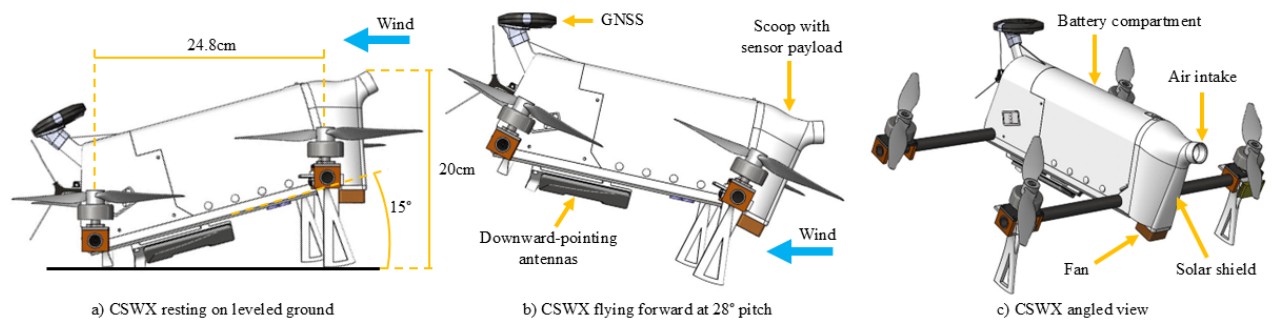

**Figure 5.** a) Right-side view of the CopterSonde-SWX (CSWX) resting on level ground and typical orientation during hover conditions in calm wind. Key dimensions are provided for scale. The wind-direction arrow is included for reference, clearly indicating the front and rear of the platform and highlighting the typical flight heading of the CSWX during vertical atmospheric profiling. b) Side view of the CSWX in forward-flight configuration, exhibiting a 28° pitch angle to counteract wind-induced drag. In this configuration, the aerodynamic profile of the CSWX effectively reduces air resistance, which is the primary purpose of the inclined body design. External components, such as the GNSS antenna and air intake, align with the wind vector, further enhancing aerodynamic efficiency. c) Angled perspective view of the CSWX, providing a three-dimensional visualization of the UAS and its structural components.

directly below their corresponding motor-mounted propellers, leveraging the propeller-induced airflow. Each propeller has a size of $10 \times 4.5$ in and can produce a maximum thrust of approximately 3.8 kgf when spinning at $14,000$ rpm.

Table 1 summarizes the CSWX's measured technical specifications based on extensive flight testing, with values in parentheses representing theoretical maximum capabilities achievable under the right conditions, even if they were not reached during testing.

## 3.2 Operating Modes

The CSWX was intentionally designed to be modular and scalable. Unlike the CS3D, the battery compartment can accommodate various battery types and capacities, provided they share similar voltage characteristics. Leveraging this design feature, two distinct operating modes were developed to showcase the scalability and adaptability of the CSWX based on battery selection.

In its highest-performance configuration, the CSWX utilizes a $6,000$ mAh (130 Wh) 6S LiPo battery capable of delivering a maximum continuous discharge current of $168$ A, evenly distributed among the four rotors. This powerful setup enables the CSWX to operate reliably under extreme conditions, allowing it to power through wind layers with quasi-horizontal speeds up to a theoretical maximum of $\sim 30$ m s$^{-1}$ before automatically activating its return-to-launch failsafe mechanism. In this configuration, the maximum attainable altitude—generally $< 1,500$ m above ground level (AGL)—is strongly influenced by power consumption resulting from wind resistance. The specifications provided in Table 1 are derived based on this high-power operating mode.





Alternatively, a second operational mode employs a 9000 mAh (194 Wh) 6S LiIon battery capable of providing a maximum discharge current of 90 A. Although approximately 10% heavier than the LiPo battery, this LiIon battery stores 50% more energy, resulting in approximately 30% longer flight endurance. This increased endurance allows the CSWX to achieve higher altitudes—potentially exceeding 2,000 m AGL—which requires beyond visual line of sight (BVLOS) authorization. However, this operating mode necessitates flying in low-wind conditions to ensure optimal performance.

Overall, this flexibility significantly broadens the CSWX's operational envelope over previous CopterSonde generations, enabling a wider range of atmospheric research applications and facilitating detailed studies into the design limitations and potential enhancements of weather-sensing UAS technology.

## 3.3 Autopilot System

The CSWX is controlled using a CubePilot CubeOrange autopilot running a customized ArduPilot firmware. This firmware integrates the advanced flight and sensor features originally developed for the CS3D and extends them for the CSWX. Custom firmware incorporates and enhances key capabilities from the CS3D—such as automated waypoint mission planning, environment-aware failsafe logic (e.g., wind limits), our wind vane flight mode (WVFM), and compact custom telemetry messages (Segales et al., 2024). For communications, the Android-based HereLink $2.4$ GHz radio system is used, which provides a robust, real-time link—up to around 20 km—to the ground control station (GCS). At the GCS, we use enhanced visualization software tailored to display and analyze CSWX telemetry and sensor data, as detailed in a later section.

## 3.4 Enhanced Weather Oriented Features

In addition to the selected high-power propulsion configuration, we incorporated and tested several new and enhanced features into the CSWX that benefit weather sampling. These features were designed to improve the sampling of air from the PBL and boost the accuracy of thermodynamic and kinematic measurements. Concurrently, enhancing high-wind resilience was also a priority; the CSWX's internal components were repackaged and rearranged to create a more compact layout and streamline its longitudinal aerodynamic profile to achieve lower drag. A detailed description of these enhancements is provided in the following subsections, along with supporting evidence demonstrating their effectiveness.

### 3.4.1 Thermodynamic Sensor Suite

The front shell—comprising the scoop and payload compartment—was shaped and integrated with the CSWX's outer shell to produce a streamlined aerodynamic enclosure. The front shell is fully detachable to facilitate the installation of different sensor package configurations, as well as calibration and maintenance tasks. The CSWX's front shell is equipped with multiple thermohygrometers for measurement redundancy: three iMet-XF bead thermistors and three IST HYT-271 capacitance humidity sensors. Following the University of Oklahoma's patented sensor placement in Chilson et al. (2021) for WxUAS, the sensors were arranged in an inverted "V" shape within a cylindrical chamber.





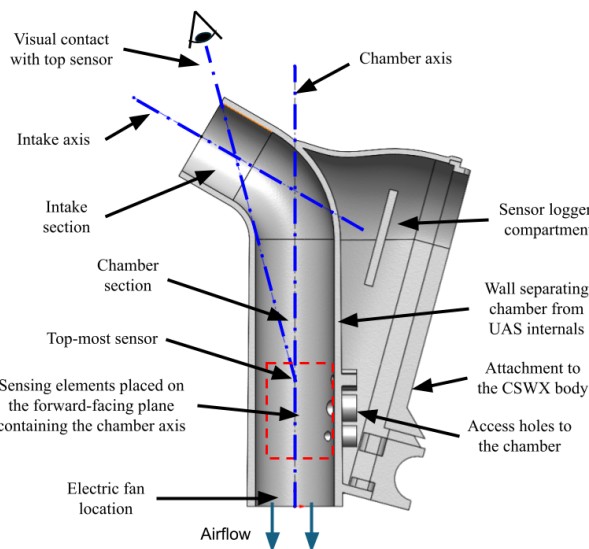

**Figure 6.** Sectional view of the CSWX front shell showing the airflow across the scoop and internal sensor chamber. Sensors are arranged inside the red dashed box with their sensing elements placed on the forward-facing plane containing the chamber's axis. The scoop's geometry and fan placement were specifically designed to ensure airflow aspiration while blocking direct line-of-sight (visual contact) to the sensing elements. The inlet curvature, chamber diameter, and sensor depth combine to protect the thermohygrometers from solar radiation and external disturbances.

Figure 6 illustrates a sectional view of the CSWX front shell showing the airflow across the scoop and internal sensor chamber. The scoop's geometry and fan placement were designed to ensure airflow aspiration while blocking direct line-of-sight to the sensing elements. The inlet curvature, chamber diameter, and sensor depth combine to protect the thermohygrometers from
solar radiation regardless of solar incidence angle.

To quantify the influence of solar heating on onboard thermohygrometer measurements, we conducted computational flow-simulation studies in SolidWorks® followed by a comparison against field data from both the CS3D and CSWX platforms. In the flow simulations, each run replicated similar conditions (solar incident angle, ambient temperature, wind speed, drone pitch, cloud cover, etc.) recorded during corresponding flight days. Both CopterSonde scoops (CS3D and CSWX) were CAD-
modeled with all three temperature and three humidity sensors, including the fan for aspiration. These experiments helped establish a baseline distribution of airflow and temperature inside each scoop under purely aerodynamic and solar effects, allowing us to isolate and measure the contributions of solar radiation and aerodynamics to the measurement bias at varying wind speeds and sun angles.

Flow simulations offered our first glimpse into how sunlight and ambient wind interact with the CSWX sensor housing.
As shown in Fig. 7, sunlit areas on the front shell heat up noticeably, with a clear hot spot forming just behind and below the intake, but not reaching the weather sensors located deeper in the scoop. This shows that shielding the instruments from direct and reflected solar radiation is critical to prevent measurement bias, but it also shows that any cover may introduce



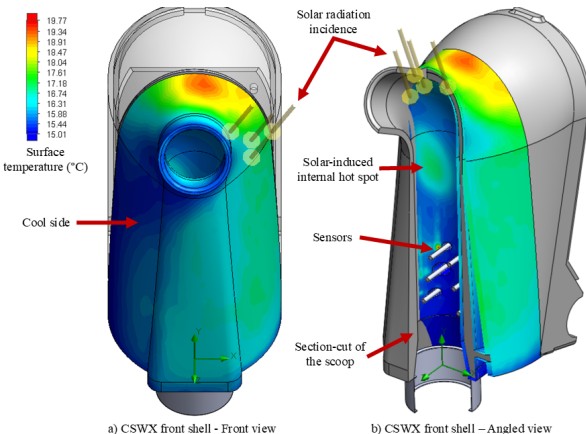

**Figure 7.** SolidWorks® FlowSim–derived surface-temperature heatmaps of the CSWX front shell under a representative solar incidence angle of $45°$ (solar irradiance = 800 $Wm^{-2}$, ambient = $15°C$, wind = 2 m $s^{-1}$). Left image (panel a)): front view of the CSWX scoop exterior, with cooler regions (blue hues) indicating surfaces shaded from direct sun and warmer regions (yellow-to-red hues) showing panels exposed to solar irradiance. Right image (panel b)): Section-cut through the scoop wall revealing the internal wall-temperature distribution. A localized hot spot appears just below the intake (red region, $17°C$), indicating a potential bias risk for this particular solar incidence. This hotspot shows the importance of placing the thermohygrometer deep inside the scoop or even applying internal reflective coatings to prevent radiative heating of the sensing elements.

other sources of error, since heated walls disrupt local airflow and introduce convective heating. In practice, the final sensor enclosure must therefore have a balance between sufficient sun protection and adequate ventilation, all while maintaining a

smooth, aerodynamic profile that blends with the aircraft's body and does not compromise the UAS's flight performance.

Figure 8 illustrates the absolute temperature spread among the three onboard thermistors for both CopterSonde configurations (CSWX in red, CS3D in blue) alongside the CSWX simulation baseline (green) across different flight days (D1–D8). On days with negligible solar irradiance and moderate winds (D4 and D7), the CSWX's measured spread collapses to near or even below the simulated baseline (green), demonstrating excellent thermal uniformity; by contrast, the CS3D continues to exhibit

some variability (blue), suggesting inherent design-induced biases unrelated to solar radiation. More broadly, for the CSWX, variations in wind speed—even up to $\sim$12m $s^{-1}$—produce only marginal changes in temperature spread, indicating that convective heat transfer (enclosure to air) diminishes, homogenizing sensor readings once airflow exceeds a minimal threshold (also seen in the thermal comparison from the low-to-high-wind flights in Fig. 12). In contrast, full exposures to solar radiation (clear-sky panels) correlate with pronounced increases in both CSWX and CS3D spreads, with the CS3D showing the largest

absolute deviations. These observations confirm that solar loading is the dominant driver of sensor-to-sensor temperature discrepancies on both platforms, whereas wind speed plays a secondary role. The CSWX mitigates both effects more effectively than the CS3D.



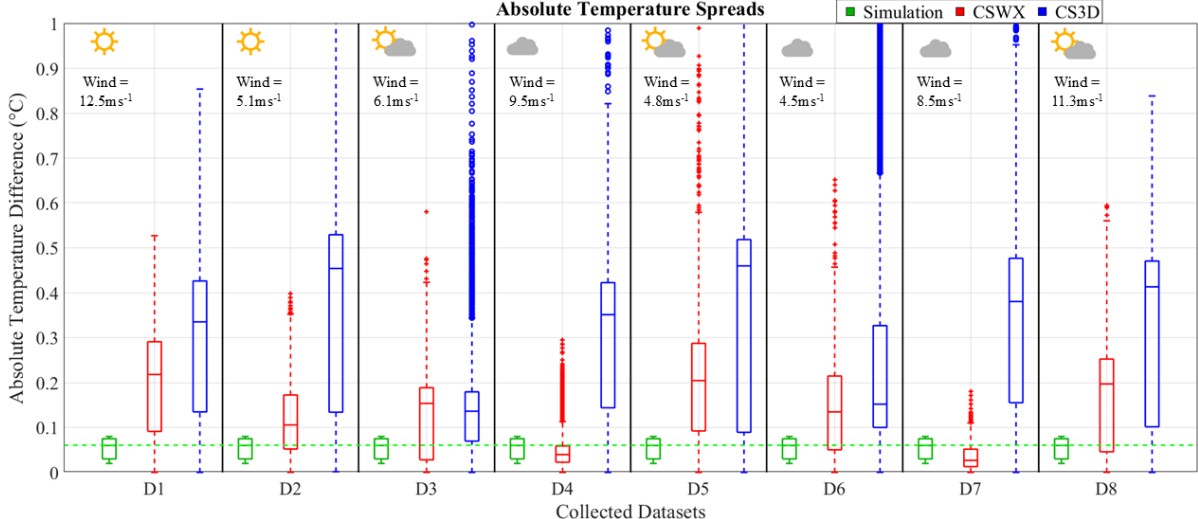

**Figure 8.** Box-and-whisker plots of absolute temperature differences between onboard thermohygrometers for eight separate sampling days (panels D1–D8). In each panel, three boxplots compare: the baseline CopterSonde-SWX CAD model simulation (green), CopterSonde-SWX field data (red), and CopterSonde-3D field data (blue). Each boxplot distribution was generated by combining 4-5 vertical profiles collected each day during Summer and Fall 2024 in Oklahoma. For the simulation, we approximated the environmental conditions on a sampling day, including sun incidence and ambient temperature. The icons above each panel indicate clear sky, scattered clouds, or overcast conditions in addition to the mean wind speed. The mean difference values between the real and simulated CopterSonde-SWX data from D1-D8 are as follows: $0.15\,°\mathrm{C}$, $0.06\,°\mathrm{C}$, $0.07\,°\mathrm{C}$, $-0.01\,°\mathrm{C}$, $0.15\,°\mathrm{C}$, $0.08\,°\mathrm{C}$, $-0.02\,°\mathrm{C}$, $0.11\,°\mathrm{C}$.

### 3.4.2 Wind Vector Estimation

Over the past five years, our team has developed and refined several algorithms to infer the wind vector using only the Copter-Sonde's flight data, since no dedicated anemometer is present onboard. These methods exploit the quadcopter's dynamics and kinematics—measured by the autopilot's inertial measurement unit (IMU)—to estimate wind vectors. The most commonly used techniques for the CS3D were based on the linear (Neumann and Bartholmai, 2015) and quadratic regression curves (Greene et al., 2022) as a function of the wind-induced tilt of the platform, assuming horizontally-steady vertical flights. These algorithms were initially implemented and evaluated on the CS3D (Bell et al., 2020) and then ported to the CSWX with the parameters retuned in open-field tests against meteorological towers and Doppler wind lidars (Segales et al., 2020; Segales, 2022).

However, the linear and quadratic methods were limited to horizontal wind estimation. To resolve the full wind vector—including the vertical component—we must leverage rotor thrust information. By accurately measuring each rotor's angular velocity and mapping it to thrust (through propeller characterization), the CopterSonde can infer the drag force acting on its airframe. Combining the known weight vector, estimated thrust, and measured accelerations into the dynamic model allows for the computation of the complete wind vector. The formulation of these relationships in a simplified linear time-invariant



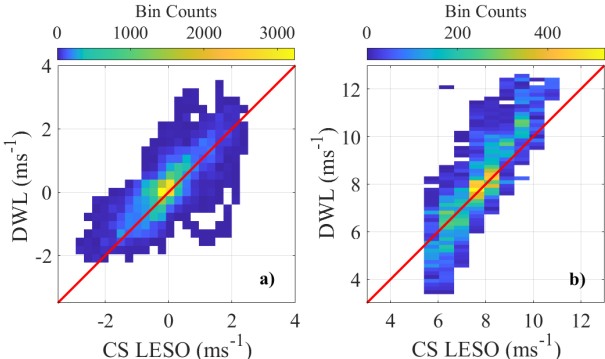

**Figure 9.** Cumulative scatter plots compare LESO-based CopterSonde (CS) vertical wind (panel a)) and horizontal wind (panel b)) estimates against Doppler wind Lidar (DWL) measurements, aggregating data from 16 vertical profiles (N = 31,386 data points). The color density in each plot indicates the frequency of observations, and a red one-to-one line marks perfect agreement. Prior to analysis, outliers from low DWL signal-to-noise ratios were removed. The correlation coefficient (R), mean difference (MD), standard deviation (SD), and RMSE for each panel are: a) vertical wind with $R = 0.791$, $MD = -0.197 \text{m s}^{-1}$, $RMSE = 0.492 \text{m s}^{-1}$, and $SD = 0.451 \text{m s}^{-1}$; panel b) horizontal wind speed with $R = 0.90$, $MD = -0.287 \text{m s}^{-1}$, $RMSE = 1.025 \text{m s}^{-1}$, and $SD = 0.984 \text{m s}^{-1}$.

(LTI) dynamic model is as follows:

$$
\begin{cases}
\dot{\boldsymbol{P}} = \boldsymbol{V} \\
\dot{\boldsymbol{V}} = \boldsymbol{g} + \frac{1}{m} R_B^I \boldsymbol{T} + \frac{1}{m} \boldsymbol{D}
\end{cases}
\Rightarrow
\begin{cases}
\dot{\mathbf{x}} = A\mathbf{x} + B\mathbf{u} \\
\mathbf{y} = C\mathbf{x}
\end{cases}, \tag{1}
$$

where $\boldsymbol{P}$ is position, $\boldsymbol{V}$ velocity, $\boldsymbol{T}$ total thrust, $\boldsymbol{D}$ aerodynamic drag, $R_B^I$ is the body-to-inertial rotation matrix, $m\boldsymbol{g}$ weight

(mass times gravity) of the quadcopter UAS. In the LTI form, $\mathbf{x}$ is the state vector, $\mathbf{u}$ the input vector, and $\mathbf{y}$ the measured outputs, with system matrices $A$, $B$, and $C$ being the state, input, and output weights, respectively. Segales (2022) shows a more detailed deduction and formulation of the presented equations.

A critical requirement for this method is the use of an accurate propeller model, for which we adopted the characterization provided by Gill and D'Andrea (2019). However, their thrust formulation depends on the inflow velocity to the propellers

(perpendicular to the rotor disk), which is equal to the airspeed of the CSWX. The airspeed of the CSWX is the difference between the CSWX ground velocity vector and the wind vector; the latter is unknown, and we aim to measure it in the first place. Therefore, we opted to use a Linear Extended State Observer (LESO) for the estimation of non-measurable parameters (Han, 2009; Li et al., 2012). In Eqn. (1), the unknown parameter is the aerodynamic drag $\boldsymbol{D}$. Subsequently, the LESO extends the state-space variables $\mathbf{x}$ to include an estimation process for $\boldsymbol{D}$ as an integrator. After applying these control-theory operations





to Eqn. (1), the resulting LTI+LESO system is:

$$
\begin{cases}
\dfrac{d\hat{\mathbf{x}}}{dt} = (A - LC)\hat{\mathbf{x}} + B_a \begin{bmatrix} u \\ h \end{bmatrix} + L\mathbf{y} \\[2em]
\mathbf{y} = C\hat{\mathbf{x}}
\end{cases}
, \tag{2}
$$

where $\hat{\mathbf{x}} = [P; V; D]^T$ is the state estimate, $u$ is the rotated thrust vector combined with the weight vector, and $h$ is the rate of change of the wind. Assuming that the mean wind changes slowly over time, then $h = 0$. The LESO then initializes with a first guess of wind velocity estimate (usually set to zero) and iteratively refines it using the wind and thrust estimate computed

in the previous step. With sufficiently high sampling rates ($\sim 10$ Hz), this approach quickly corrects and converges to stable drag estimates. The matrices in Eqn. (2) are defined as:

$$
A = \begin{bmatrix} 0 & 1 & 0 \\ 0 & 0 & \frac{1}{m} \\ 0 & 0 & 0 \end{bmatrix}, B_a = \begin{bmatrix} 0 & 0 \\ 1 & 0 \\ 0 & 1 \end{bmatrix}, \tag{3}
$$

$$
C = \begin{bmatrix} 1 & 0 & 0 \\ 0 & 1 & 0 \end{bmatrix}, \text{ and } L = \begin{bmatrix} \beta_{p1} & \beta_{v1} \\ \beta_{p2} & \beta_{v2} \\ \beta_{p3} & \beta_{v3} \end{bmatrix}. \tag{4}
$$

The observer gain $L$ determines the convergence speed and noise sensitivity; this is a user-defined parameter chosen so that

the eigenvalues of $(A - L, C)$ lie in the left half of the complex plane, ensuring system stability and estimation errors decay to zero (Stevens et al., 2015). The CSWX is capable of sampling the input parameters for the LESO estimation at a rate of 10 Hz

Figure 9 demonstrates the LESO-based CopterSonde's wind-sensing performance compared to data collected with a Doppler wind Lidar (DWL) system. The cumulative scatter plots aggregate over $31,000$ data points from 16 profiles, revealing strong correlations (R $= 0.79$ vertical, R $= 0.90$ horizontal) and low RMSEs ($0.49$m s$^{-1}$ vertical, $1.03$m s$^{-1}$ horizontal) between

LESO estimates and DWL measurements. Although both systems were tested over a relatively narrow range of horizontal wind speeds, the LESO begins to underestimate at higher wind speeds (see the green trend in Fig.9b)—likely because the simplified dynamic model underperforms in fully capturing the platform's nonlinear aerodynamic behavior at high speeds.

### 3.4.3   Angled airframe

To achieve higher resilience to high wind conditions, a powerful propulsion system alone is not enough. The airframe must

also be designed to minimize wind-induced drag at high wind speeds without sacrificing sensitivity for wind estimation under calmer conditions. Tilting or angling the UAS body relative to its rotor disk can significantly enhance high-speed flight performance by reducing aerodynamic drag. Traditional quadcopters typically tilt their entire body to produce a desired inclination of the thrust vector to compensate for wind effects or shift the UAS position in 3D space. This inclination increases the exposed frontal area, drag, and reduces aerodynamic efficiency. In contrast, a built-in tilted-body quadcopter can increase maximum




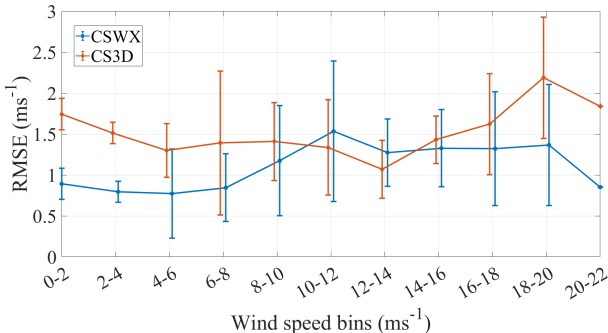

**Figure 10.** Comparison of wind estimation RMSE for the CSWX (blue) and CS3D (orange) across wind-speed bins up to 22m s$^{-1}$, including error bars of 2 standard deviation. The CSWX consistently shows lower RMSE in the 0–8 m s$^{-1}$ range, matches CS3D performance in moderate winds (8–16 m s$^{-1}$), and diverges with significantly lower errors at higher speeds, demonstrating the aerodynamic advantages of the SWX design.

achievable speeds as much as 12.5%, as demonstrated by Tang et al. (2022). However, Tang et al. (2022) recommends limiting the airframe angle to below 20° to preserve flight stability and maneuverability. Based on this guidance, the CSWX airframe was designed with a fixed airframe angle of 15°, leaving a safety margin with the upper limit.

To quantify the CSWX's wind sensitivity over the CS3D, we performed 10 side-by-side vertical profiles (up to 1,000 m) colocated with NSSL's Collaborative Lower Atmosphere Profiling System (CLAMPS) DWL reference system (Wagner et al., 295  2019), sensing wind speeds up to 22 m s$^{-1}$. Subsequently, using this dataset, we tuned our three wind estimation algorithms—linear, quadratic, and LESO—using a Differential Evolution optimizer (DEO) to minimize root-mean-square error (RMSE) relative to the DWL observations. For each CopterSonde version, RMSE values from all the methods were averaged over 2 m s$^{-1}$ wind-speed bins to produce the comparison plot in Fig. 10. Averaging across all methods ensures a fair comparison of the aircraft's aerodynamic performance independent of the performance of any single wind-estimation algorithm. The 300  results show that the CSWX outperforms the CS3D at low wind speeds (demonstrating enhanced sensitivity), matches performance at mid-range winds, and again delivers lower RMSE at higher speeds, evidencing the benefits of its refined aerodynamic design.

Figure 11 presents two examples of colocated vertical soundings from CopterSonde (CS3D and CSWX), radiosonde, and DWL. In the high-wind profile (left), all instruments align closely, with the CSWX maintaining the best accuracy at the upper 305  end of wind speeds. In the low-wind profile (right), the CSWX's LESO-based estimates closely track the references, benefiting from the LESO algorithm's ability to isolate wind-induced aerodynamic effects and the tilted airframe's enhanced sensitivity to gentle breezes.





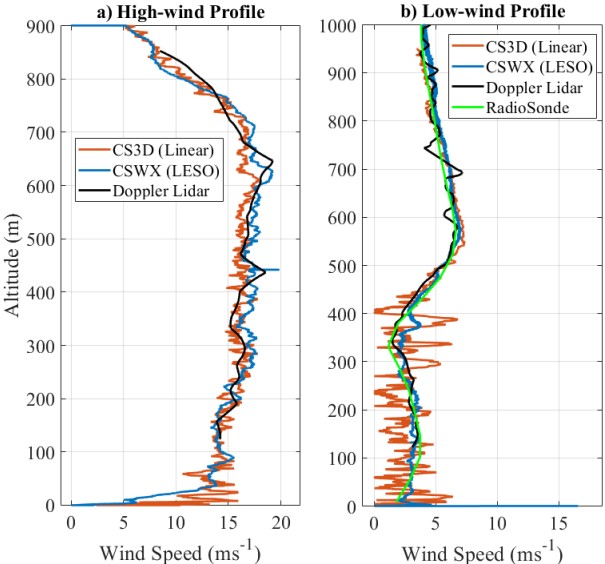

**Figure 11.** Comparison of vertical wind profiles from the CSWX (blue), CS3D (orange), Doppler wind lidar (black), and radiosonde (green) for two colocated soundings. In the high-wind profile (panel a)), all sensors align closely—CSWX most accurately matches the lidar above $15 \text{ m s}^{-1}$—while in the low-wind profile (panel b)), the CSWX's LESO-based estimates (blue) closely track both lidar and radiosonde below $6 \text{ m s}^{-1}$, whereas the CS3D (orange) shows greater scatter, highlighting the SWX's enhanced sensitivity to light winds.

## 4 Field Operations and Observations

The CopterSonde system, in its current iteration, has adhered to all relevant FAA and NOAA Office of Marine and Aviation
Operations (OMAO) regulations governing UAS operations, even during NOAA-related UAS missions. All flights conducted during this study took place at the Kessler Atmospheric and Ecological Field Station (KAEFS) in Purcell, Oklahoma, USA, located 30 km southwest of the OU Norman campus. These flights were compliant under the Certificate of Authorization (COA) with number 2024-CSA-14684-COA, issued by the Federal Aviation Administration (FAA), which allowed us to fly UAS above 400 ft AGL with a flight ceiling of $5,000$ ft AGL. Every person involved in CopterSonde missions received proper training for
safe and successful deployments in the field. The following sections detail some relevant CopterSonde deployments in which we deliberately pushed the platform toward its performance limits to evaluate its capabilities and resilience for atmospheric data collection and to make practical conclusions about its future operational use.

### 4.1 Thermodynamic and Kinematic Measurement Performance

This section focuses on presenting statistical evidence and justifying the weather-oriented engineering developments (see
Sec. 3.4) implemented on the CopterSonde-SWX during real profiling operations. The CS3D was used as the baseline from which the CSWX's measurement performance is compared. We present four case studies of side-by-side CSWX and CS3D soundings colocated with two DWLs and two Vaisala RS41-SGP radiosonde (RS) launches. Thus, we have kinematic datasets





for all four soundings from both DWL and RS but only two thermodynamic datasets from the RS. The experiment took place soon after sunrise to capture the residual nocturnal inversion so that sensor response would have an influence on the results

for a fair comparison. During the measurements of these profiles, the two DWLs were running simultaneously; one was set to fixed-point (DLFP) mode, ideal for vertical wind measurements, while the second was set to velocity-azimuth display (VAD) mode, optimized for horizontal wind measurements. Together, the DWLs took 3D-wind profiles of the PBL up to $\sim 1,500$ m AGL every 5 sec. The CSWX, CS3D, and RS were launched nearly simultaneously from locations less than 20 m apart. The climb rate for both CSWX and CS3D was set to $3.5$ m s$^{-1}$, whereas the RS balloons' pressure was calibrated such that its

ascent rate was the standard $5$ m s$^{-1}$.

    For the thermodynamic comparison, Fig. 12 shows two vertical soundings up to $1,000$ m with their respective temperature and relative humidity profiles, as well as the wind profile for a better understanding of the atmospheric conditions. These two examples exhibit a strong thermal inversion, including a well-defined shear layer at approximately $500$ m AGL, useful for identifying and comparing measurement effectiveness related to aspiration and sensor response. Under higher winds, all

instruments agree closely; in light winds, the CSWX exhibits notably tighter alignment, smoother and more consistent temperature and relative humidity curves, confirming that its enhanced aspiration, solar shielding, and tilted-body design effectively mitigate self-heating and radiation biases.

    Table 2 presents a statistical assessment of thermodynamic measurement performance based on the presented vertical profiles. The table reports the root-mean-square error (RMSE) of each UAS platform's profile relative to the RS data. The results

show that the CSWX achieved lower RMSE values than the CS3D, indicating its thermodynamic outperformance over its predecessor through engineering improvements as described in Sec. 3.4.1. Assuming that the RMSE reflects the CSWX's estimation accuracy, doubling those values gives 95% confidence intervals of $\pm 0.78°$C for temperature and $\pm 6.62\%$ for relative humidity.

    Figure 13 shows profiles of the horizontal and vertical wind where both CSWX and DWL captured wind shear, down-

drafts, updrafts, and regions of vertically steady flow. These plots demonstrate that the LESO's estimates of horizontal and vertical wind closely followed the DWL's observed trends within an acceptable error envelope. Table 3 provides a quantitative evaluation of each CopterSonde's wind-estimation performance, reporting the RMSE of their profiles relative to DWL data (VAD and DLFP) and, where available, RS measurements. The CSWX consistently achieved lower RMSE values than the CS3D—evidence of its engineered improvements—and its mean RMSE for both horizontal and vertical winds remained

below $1$ m s$^{-1}$. These results indicate that a multirotor UAS, when aerodynamically and structurally optimized for atmospheric sampling, can deliver wind measurements comparable to those from conventional instruments without requiring onboard wind-specific sensors. In contrast to temperature and relative humidity, treating the wind RMSE as a direct estimation of the CSWX's true accuracy can be misleading. Differences in sensor response and sampling rate between the CSWX, DWL, and RS may have been significant such that the CSWX captured smaller-scale wind fluctuations that appear as large devi-

ations from the DWL and RS, thereby inflating the RMSE (a metric that heavily penalizes large differences). Accordingly, wind speed and direction RMSE values should be interpreted with caution and not assumed to represent the CSWX's inherent wind-measurement accuracy.





**Table 2.** Root mean squared error (RMSE) of both CopterSonde (CS)—CSWX and CS3D—temperature and relative humidity profiles with respect to Vaisala radiosonde (RS) profiles (see Fig. 12). The instruments had a horizontal spatial separation of less than 20 m between each other and were launched almost simultaneously.

| RMSE of CS w.r.t. RS | Temperature | Rel. Humidity | Wind speed |
|---|---|---|---|
| CopterSonde-SWX (LESO-based wind algorithm) | | | |
| Profile 2 | 0.27°C | 1.58% | 0.48 m s$^{-1}$ |
| Profile 4 | 0.50°C | 2.71% | 1.00 m s$^{-1}$ |
| Average | 0.39°C | 2.15% | 0.74 m s$^{-1}$ |
| CopterSonde-3D (Quadratic-based wind algorithm) | | | |
| Profile 2 | 0.41°C | 2.11% | 1.18 m s$^{-1}$ |
| Profile 4 | 0.70°C | 3.56% | 1.30 m s$^{-1}$ |
| Average | 0.56°C | 2.83% | 1.24 m s$^{-1}$ |

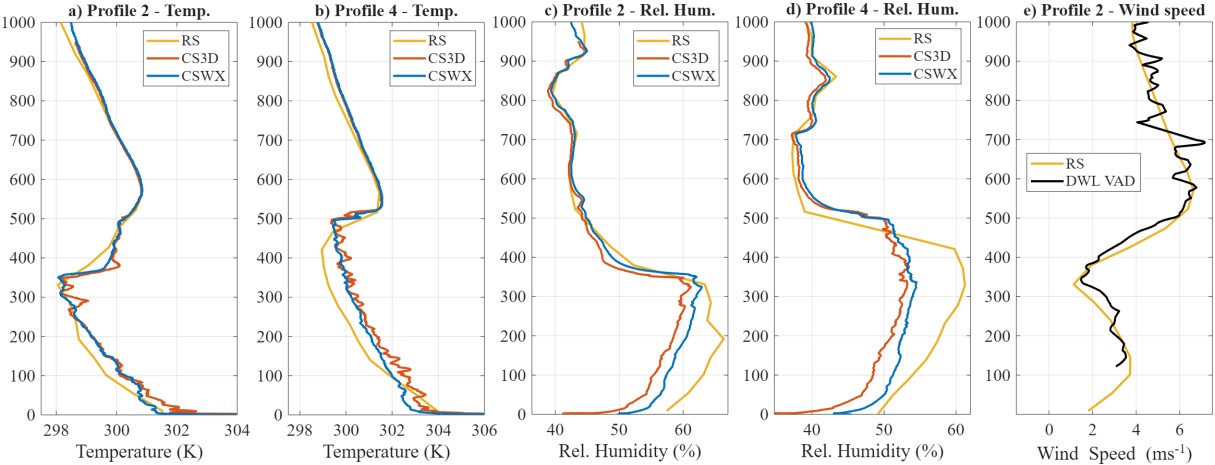

**Figure 12.** Profiles of temperature and relative humidity from two CopterSonde models (SWX in blue, CS3D in red) alongside radiosonde (RS) in orange and Doppler wind lidar (DWL) references in black. The first four panels show temperature (panels a) and b)) and relative humidity (panels c) and d)) for profiles 2 and 4 up to 1000 m. Panel e overlays wind speed, highlighting low ($< 4$ m s$^{-1}$) and high ($> 4$ m s$^{-1}$) wind layers near 450 m. Under stronger winds, all instruments converge closely, whereas light winds produce greater scatter, demonstrating that the SWX's improved aspiration and shielding effectively suppress sensor self-heating and solar-radiation biases, as evidenced by its smoother low-wind temperature traces.




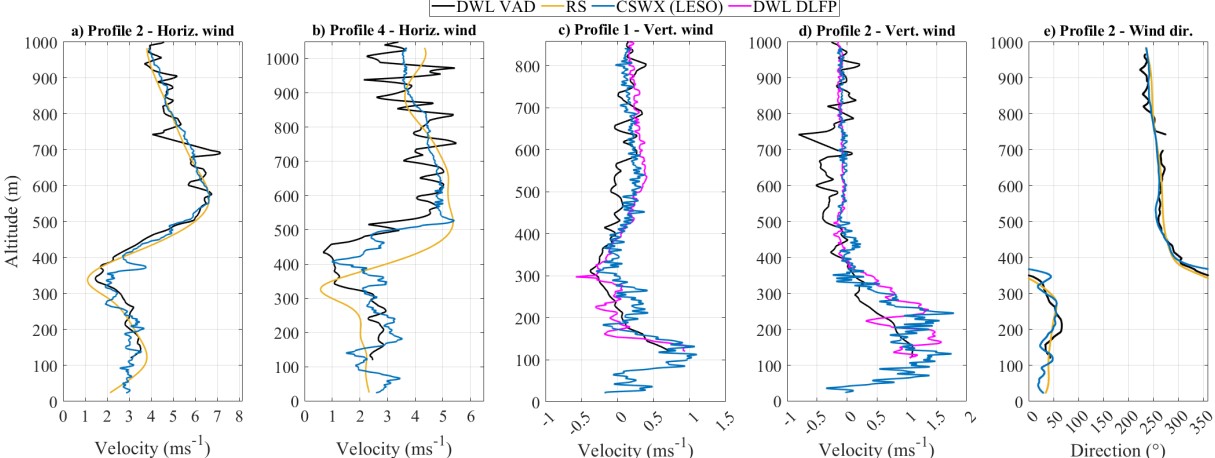

**Figure 13.** Vertical profile comparison between LESO-based CopterSonde-SWX (CSWX) and two Doppler wind lidars (DWL); one set in fixed-point (DLFP) mode and the other in vertical-azimuth display (VAD) mode, showing two different profiles of horizontal wind speed (panels a) and b)), vertical wind velocity (panels c) and d)), and wind direction (panel e)). The CSWX collected samples every 0.1 sec, while both DWLs took samples every 5 sec. The spatial separation between the CSWX and both DWLs was 20 m. These profiles were produced from the same CSWX flights and DWL measurements shown in Fig. 12.

## 4.2 Severe Weather Deployment

In this section, we outline our approach for deploying the CSWX in challenging PBL conditions. We describe the operational
strategies, measurement objectives, and evaluation metrics used to assess its wind tolerance, precipitation resilience, and data-quality performance. Through targeted case studies and systematic analyses, we demonstrate how the CSWX enables profiling of thermodynamic and kinematic structures in environments where conventional UAS often fail. Although the CSWX's heavy power consumption reduces flight time, it grants us the unique opportunity to characterize UAS performance under extreme conditions. Additionally, this trade-off would enable measurements of fast-evolving thermodynamic and kinematic features in
atmospheric conditions almost entirely unexplored by WxUAS.

To define the CSWX's propulsion limits and develop real-time failsafe algorithms, we conducted a series of controlled stress flights in which we logged instantaneous voltage, current, power, and airspeed across a range of extreme aerial maneuvers and high-speed test flights. Additionally, the CSWX collected vertical profiles on windy days, which extended the propulsion characterization to realistic ascent–descent profiles. For all vertical profiles, both CopterSondes were programmed to ascend at
$3.5 \text{ m s}^{-1}$ and descend at $5 \text{ m s}^{-1}$. From these experiments, we derived third-order polynomial expressions for the ascent-leg power $P_a$ and descent-leg power $P_d$ consumptions of the CSWX as functions of horizontal wind speed $U$ (in m s$^{-1}$), yielding:

$$P_a(U) = 0.0753U^3 - 1.344U^2 + 15.32U + 419 \,, \tag{5}$$
$$P_d(U) = 0.0963U^3 - 2.63U^2 + 12.14U + 352 \,, \tag{6}$$





**Table 3.** Root mean squared error (RMSE) of both CopterSonde (CS)—CSWX and CS3D—wind profiles relative to two Doppler wind lidar (DWL) and Vaisala radiosonde (RS) profiles (see Fig. 13). One DWL was set to Fixed-point (DLFP) mode, ideal for vertical wind measurements, while the second was set to vertical-azimuth display (VAD) mode, optimized for horizontal wind measurements. The instruments had a horizontal spatial separation of less than 20 m between each other and were launched almost simultaneously.

| RMSE of CS w.r.t. DWL and RS | VAD | RS | VAD | RS | DLFP |
|---|---|---|---|---|---|
| | Horizontal ($\mathrm{m\ s^{-1}}$) | | Direction (°) | | Vertical ($\mathrm{m\ s^{-1}}$) |
| CopterSonde-SWX (LESO-based wind algorithm) | | | | | |
| Profile 1 | 0.56 | — | 8.31 | — | 0.20 |
| Profile 2 | 0.51 | 0.48 | 8.21 | 8.17 | 0.26 |
| Profile 3 | 0.71 | — | 10.27 | — | 0.25 |
| Profile 4 | 0.75 | 1.00 | 11.68 | 6.88 | 0.28 |
| Average | 0.63 | 0.74 | 9.62 | 7.52 | 0.25 |
| CopterSonde-3D (Quadratic-based wind algorithm) | | | | | |
| Profile 1 | 1.01 | — | 7.32 | — | — |
| Profile 2 | 1.10 | 1.18 | 8.87 | 9.34 | — |
| Profile 3 | 1.30 | — | 8.15 | — | — |
| Profile 4 | 1.27 | 1.30 | 12.75 | 8.35 | — |
| Average | 1.17 | 1.24 | 9.27 | 8.85 | — |

with power in W. These relationships were used in the following section to quantify the CSWX's energy consumption under
varying wind conditions and form the foundation for autonomous and real-time protection against propulsion overstress.

### 4.2.1   Flights in High Wind

Over the last five years and through numerous field experiments as referenced in Sec. 3, the CS3D has been thoroughly characterized, enabling us to consider its maximum safe wind tolerance of $20\ \mathrm{m\ s^{-1}}$ as accurately established. Figure 14 presents a case where the CSWX and CS3D, both colocated and simultaneously, measured wind profiles of a nocturnal low-level jet
(LLJ). In panel a), wind estimates from each UAS show that the CS3D reached its wind limit of $20\ \mathrm{m\ s^{-1}}$ at approximately 275 m, automatically triggering return-to-launch (RTL) mode, whereas the CSWX sustained winds approaching $24\ \mathrm{m\ s^{-1}}$ up to about 520 m.

Given that the CSWX was a less-well-characterized platform, its ascent was terminated by operator intervention for caution rather than by reaching an unproven estimated wind threshold. Figure 14, panel b), shows the instantaneous throttle values
ranging from 0 to 1 for each platform during ascent. Maximum available throttle ($Th_M$) is internally calculated by the autopilot based on the current UAS attitude and available power to achieve a stable flight. The throttle out ($Th_O$) is the amount of throttle





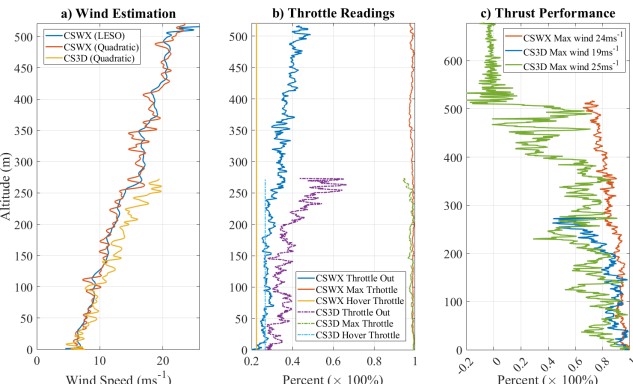

**Figure 14.** Comparison of CSWX and CS3D wind estimation and thrust performance during a colocated, simultaneous nocturnal LLJ sounding. Panel a): wind speed profiles versus altitude for CSWX (blue: LESO-based estimate; orange: linear estimate) and CS3D (yellow: quadratic estimation), showing the CS3D reaching its $20$ m s$^{-1}$ wind limit at $\sim 275$ m (triggering RTL) while the CSWX sustains winds up to $\sim 24$ m s$^{-1}$ at $\sim 520$ m before operator intervention. Panel b): Autopilot throttle metrics during ascent, such as available thrust $Th_M$, commanded throttle $Th_O$, and hover throttle $Th_H$ expressed as a fraction of maximum output. Panel c): Normalized thrust performance as formulated in Eqn. (7) plotted versus altitude, illustrating the CS3D safe cutoff at $Th_P \approx 0.4$ and CSWX's extended thrust margin under high-wind conditions. A maxed-out CS3D case is also depicted in green where $25$ m s$^{-1}$ winds exhausted its available thrust ($Th_P \approx 0$), causing the CS3D to be pushed off course.

requested by the autopilot to the rotors, while the throttle hover ($Th_H$) is the estimated throttle required to maintain a stationary hover.

To quantify thrust performance $Th_P$ of each CopterSonde, we computed the difference between $Th_M$ and $Th_O$ and then normalized it by the difference between $Th_M$ and their corresponding $Th_H$, as expressed by:

$$Th_P = \frac{Th_M - Th_O}{Th_M - Th_H} \; . \tag{7}$$

Here, the numerator $Th_M - Th_O$ represents the excess thrust available above the thrust required for the current operating condition, while the denominator $Th_M - Th_H$ quantifies the total thrust margin above the hover thrust (the minimum thrust needed to maintain flight in calm conditions). Therefore, $Th_P$ becomes a dimensionless metric of how much of the available thrust reserve is still usable to counter external disturbances, allowing for consistent comparison across flight regimes and platforms.

Figure 14, panel c), shows the resultant $Th_P$ for both CSWX and CS3D in red and blue, respectively, for this particular example. Once the CS3D's $Th_P$ falls to approximately $0.4$ (the predefined safe margin), it automatically initiates RTL. We therefore adopted this thrust-based safety cutoff as a high-wind performance metric to evaluate the CSWX, estimate its maximum wind tolerance, and predict the altitude at which the CSWX would have triggered RTL, assuming that the wind profile approximates a quadratic curve.





The quadratic curve fitting was used to determine $Th_P = f(h)$ and $U = f(h)$ from Fig. 14, where $h$ is the altitude above ground in meters. The resulting polynomials are:

$$Th_P(h) = -6.831 \times 10^{-7}h^2 - 1.252 \times 10^{-4}h + 0.96 \,, \tag{8}$$

$$U(h) = -1.338 \times 10^{-5}h^2 + 39.8 \times 10^{-3}h + 5.924 \,. \tag{9}$$

By setting $Th_P(h) = 0.4$ and solving for $h$ we obtained the estimated altitude at which the CSWX would have reached its maximum wind tolerance. We then applied the estimated max altitude to the wind-speed fit $U(h)$ to obtain the corresponding safe mean-wind limit. For the profile shown in Fig. 14, this procedure yielded $h_{safe} \approx 818$ m and $U_{safe} \approx 29.5$ m s$^{-1}$. The absolute mean-wind limit ($Th_P = 0$) corresponds to $U_{max} \approx 33.5$ m s$^{-1}$.

Next, we evaluated whether the CSWX's stored energy is sufficient to reach the safe altitude $h_{safe}$ under sustained high-wind conditions. The CSWX's total energy consumption $E_T$ was obtained by summing the integrals Eqns. (5) and (6) over the ascent and descent legs.

$$E_T = \int_0^{T_a} P_a(U(h(t)))dt + \int_0^{T_d} P_d(U(h(t)))dt \,, \tag{10}$$

where $T_a$ and $T_d$ are the total ascent and descent times, respectively, which can be calculated using the UAS's fixed ascent and descent speeds. For the presented theoretical flight case up to $h_{safe} = 818$ m and assuming quadratic increase of wind speed to 29.5m s$^{-1}$ from Eqn. (9), it yields $E_T \approx 82$ Wh. This is below the LiPo battery's total energy capacity of 130 Wh. Thus, we anticipate that the CSWX can safely reach 818 m and return to its launch point with roughly 35% of the total energy still available. Under the same wind conditions, this represents almost 200% increase in maximum altitude compared to the CS3D.

Unfortunately, we could not empirically validate these estimates because no sufficiently strong LLJ occurred during this study to test the CSWX's wind limits. However, achieving a top speed of 35.6m s$^{-1}$ during stress-flight tests and using the well-characterized CS3D as a proxy gave us confidence in the feasibility of these performance predictions.

### 4.2.2 Flights in Rain

At the KAEFS site, in addition to the Oklahoma Mesonet tower, we deployed an OTT Pluvio$^2$ L precipitation weighing gauge (pluviometer) (Oue et al., 2025). Because the CSWX must withstand both high winds and precipitation during severe weather, it was designed to be nearly sealed, with only a few strategically placed vents for airflow and heat dissipation. To evaluate its performance in wet conditions, we conducted flights on a rainy day at KAEFS. Overcast skies with a low cloud ceiling of approximately 700 m limited our maximum flight altitude to around 300 m due to poor visibility.

Figure 15 shows the precipitation rate recorded by the pluviometer (red line) during CSWX operations, with vertical dashed lines marking each flight. Although the gauge indicates almost no rainfall between 1900 UTC and 2100 UTC, ambient humidity was elevated. After 2100 UTC, the CSWX encountered moderate rain during its flights, which it handled successfully. Figure 16 presents the temperature and relative humidity profiles measured by the CSWX for the last four flights between approximately





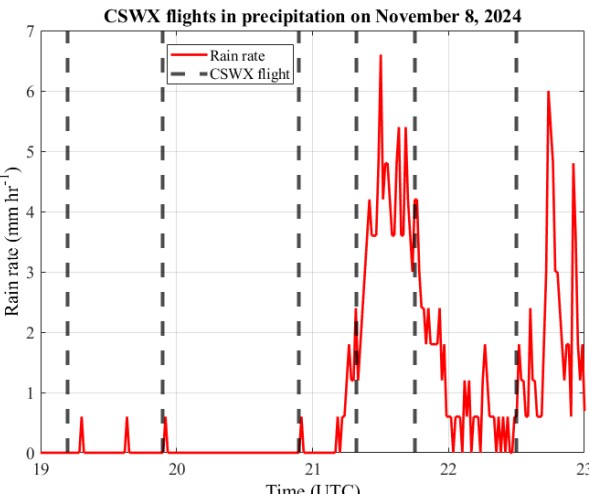

**Figure 15.** Precipitation rate measured by the OTT Pluvio$^2$ L weighing gauge at KAEFS on 8 November 2024 (red line), with vertical dashed lines indicating each CSWX flight. Light rainfall and elevated humidity preceded 2100 UTC, after which moderate rain coincided with CSWX operations up to 2300 UTC, demonstrating the platform's exposure to varying wet conditions.

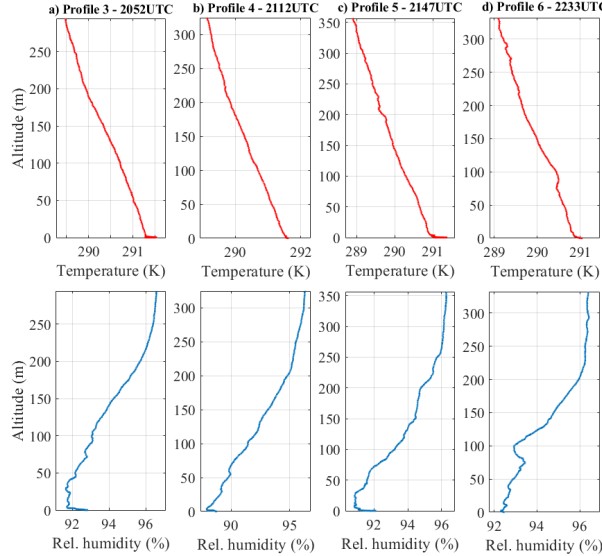

**Figure 16.** Vertical profiles of temperature (top row, red) and relative humidity (bottom row, blue) recorded by the CSWX during its last four flights between approximately 2100 UTC and 2300 UTC, profiles 3–6 from Fig. 15. The smooth, monotonic curves through altitudes up to 350 m indicate that the thermohygrometer produced accurate readings without rain-induced artifacts.

2100 UTC and 2300 UTC. These profiles confirm that the CSWX operated in highly humid conditions without exhibiting artifacts or inaccuracies in its thermohygrometer measurements under moderate rain.



## 5 Conclusions

This work has presented the CopterSonde concept, tracing its evolution from early prototypes to the severe-weather–capable CopterSonde-SWX. Extensive field campaigns under diverse weather conditions have shaped and refined the CopterSonde into a proven weather-sensing UAS, setting it apart from conventional platforms. Three core innovations distinguish the CSWX design:

- Wind-vane flight mode. A custom autopilot routine continuously aligns the weather sensor package into the ambient wind, ensuring that thermohygrometer sensors sample relatively undisturbed air even in gusty conditions.

- Shielded, active—ventilation sensor scoop. The front shell—with internal air circulation and solar shielding—combined with the wind-vane behavior, minimizes rotor-wash contamination and radiative heating, while preserving rapid aspiration through strategically placed vents.

- High-thrust, tilted-body airframe. An angled fuselage paired with a powerful propulsion system trades endurance for high-wind tolerance and aerodynamic efficiency, enabling us to characterize WxUAS in above-average harsh weather conditions.

Comprehensive calibration and field validation demonstrate that these enhancements deliver measurable improvements. In side-by-side flights with the CS3D predecessor and radiosondes, the CSWX maintained inter-sensor temperature uniformity within $\pm 0.2°C$ across diverse sun irradiance and wind regimes. LESO-based wind retrievals—validated against DWL and RS—achieved RMSE of $0.49 \text{ m s}^{-1}$ (vertical) and $1.03 \text{ m s}^{-1}$ (horizontal), outperforming earlier CopterSonde models and showing tight agreement even under light-wind conditions.

Severe-weather flights further confirm the CSWX's resilience. In a nocturnal low-level jet sounding, the CSWX sustained winds up to $24 \text{ m s}^{-1}$ at 520 m (beyond the CS3D's $20 \text{ m s}^{-1}$ at 275 m limit) before operator intervention. Polynomial fits predict a safe maximum wind tolerance of $29.5 \text{ m s}^{-1}$ and absolute wind limit of $33.5 \text{ m s}^{-1}$. An example profile showed that in high-wind conditions the CSWX could reach an altitude of 818 m before automatically triggering RTL, returning to the launch point with $\sim 35\%$ battery energy remaining. Rain-flight tests at KAEFS on an overcast and rainy evening produced smooth temperature and humidity profiles to 350 m, with no sign of rain-induced artifacts, validating the splash-resistant enclosure.

Together, these findings establish the CSWX as a robust in situ profiler capable of penetrating and characterizing extreme PBL conditions—both high-wind and wet—challenging conventional sUAS platforms. The CSWX's modular design, advanced weather-oriented autopilot features, and demonstrated performance make it a promising candidate for networked severe-weather observing systems and future data-assimilation efforts in numerical weather prediction. Ongoing and future work will focus on integrating higher-energy-density batteries, enabling BVLOS operations, and adding complementary sensors (e.g., trace gases, aerosol counters) to further broaden the CopterSonde's scientific utility.

*Data availability.* The data mentioned in the article can be obtained by contacting the corresponding author.



*Author contributions.* AS conceptualized the study. AS, TB, and AT curated the data, developed the methodology, and implemented the software. AS, AT, AQ, and JS performed the formal analysis and prepared the visualizations. AS and TB supervised, administered, and validated the project. AS, TB, JG, and ES conducted the investigation. ES acquired the funding and, together with AS and TB, provided the resources. AS, AT, and AQ wrote the original draft, and AS, TB, and JG reviewed and edited the manuscript with contributions from all co-authors.

*Competing interests.* The authors have no competing interests to declare.

*Acknowledgements.* Funding was provided by NOAA/Office of Oceanic and Atmospheric Research under NOAA-University of Oklahoma Cooperative Agreement #NA21OAR4320204, U.S. Department of Commerce.



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
