# Peer review of "CopterSonde-SWX: Development of a UAS-based Vertical Atmospheric Profiler for Severe Weather"

_EGUsphere, 2025_

## Referee Comment (RC1)

Referee Report on
**CopterSonde-SWX: Development of a UAS-based Vertical Atmospheric Profiler for Severe Weather**

The article presents an overview of the CopterSonde-SWX uncrewed aircraft system (UAS), which is an updated version of the Coptersonde 3D UAS. This version incorporates additional improvements to the battery packaging, motors, and motor arrangement to improve the UAS's ability to resist high winds (hence SWX for severe weather). Additional improvements are made to the wind extraction algorithm to allow the aircraft to recover the 3D wind vector. After reviewing the genesis of the Coptersonde, the updates in this latest version are described, and the results of intercomparison experiments are presented.

In general, the article is well written, and to the extent that the authors analysis is conducted, appears to be appropriate. However, I did find that the language used in some places strayed from the technical into the effusive when describing the Coptersonde, with several instances outlined in the detailed comments below. As also described below, I found that a large portion of Section 2 detailing field experiments made by the Coptersonde 3D seemed out of place in a paper describing a different aircraft.

I would therefore recommend that the authors provided some revisions to the text to improve readability, but otherwise find that the content is suitable for publication in Atmospheric Measurement Techniques. More specific comments are:

1. The acronyms LESO and WxUAS are used in the abstract without definition. The acronym PBL is defined, but not used at any other point in the abstract.

2. Lines 108-123 include what appear to be new results from experiments conducted with the Coptersonde 3D, presumably to demonstrate that the temperature is heading dependent and the need for orientation of the aircraft into the wind to improve temperature measurement using their sensor packaging ('the scoop'). However, this section does not fully describe the experiment, lacking detailed description and is too detailed for a paper describing a different aircraft (with a different scoop design). Perhaps this section of text is better placed in an appendix, with and replaced with a statement saying something like ("Experiments with the CS3D (see Appendix A) have demonstrated the need for WVFM with this sensor placement to minimize errors which may occur when the winds are misaligned by over 50° from the aircraft's heading.")

3. Table 1 provides some specifications for the aircraft which provide both 'maximum limits' and 'theoretical maximums'. The distinction is not really made clear in the text (except for maybe the altitude above ground). It seems to me that including theoretical maximums obscures the information if the aircraft's limits are below these values. Are the maximum limits the maximum value reached during testing? Or are they really maximum limits?

4. Lines 128-137 Seems to repeat information from the introduction (motivation). Also, there is language that is 'sales pitchy' in the last sentence, for example 'high-power platform' is not specific, since there are certainly platforms with much higher power output than the Coptersonde SWX. 'UAS-hostile conditions' can encompass a wide range of conditions including GPS-denied environments, high RF environments, even low density atmospheres can be considered UAS-hostile conditions. Furthermore, I don't recall seeing anywhere in the paper where the 'robust safety diagnostics' are described in detail (unless the authors mean where they have implemented identification of high wind conditions?) and 'weather-dependent operational modes' doesn't specify which types of weather (icing? snow? rain?). Most of the modifications to Coptersonde SWX appear to be designed to improve its resistance to high winds, but there are many other weather and environmental factors which can impact UAS performance.

5. Line 140 Another sales pitchy statement 'push its performance to the limits' What are the limits? based on table 1, the aircraft cannot reach its theoretical limits under many performance criteria, so this statement feels very out of place in a scientific paper.

6. Line 148 Another vague statement that lacks evidence is 'achieve high aerodynamic performance'. The phrase high aerodynamic performance implies what in this context? I would argue that fixed-wing

UAS, which have a high lift-to-drag ratio (relative to similar-sized UAS) would merit this phrasing, but I don't know how aerodynamic performance of a rotorcraft UAS is quantified and no evidence is provided demonstrating that it is better for the Coptersonde SWX than for any other similar scale of aircraft.

7. Line 158 'Driven by a high-performance electronic speed controller' is another example where adjectives are introduced without context. What is high performance for an electronic speed controller, does 80A make it high performance? Or are there other technical capabilities of this ESC that make it perform better than equivalently rated ESCs?

8. Line 161 'propeller has a size of $10 \times 4.5$in". The 10 in the prop description describes its diameter and pitch. Not just size. Also, these dimensions should probably be metric.

9. Line 180 '-which requires beyond visual line of sight (BVLOS) authorization'. This statement should probably also require the qualifying statement that this will be jurisdiction dependent. For example flights over the ocean or in restricted airspace will not require such authorization.

10. Lines 197 and 202 The term 'streamline' and 'streamlined' lack technical definition and is somewhat vague in this context. In the first instance, do the authors mean reducing the frontal area to reduce pressure drag? Or do they mean reducing the length of the body to reduce skin friction drag? Better to be more precise in the terminology used. In the second instance, do they mean reducing step changes in the surface or surface roughness that can introduce additional flow separation and drag?

11. Section 3.4.1

    (a) This section does not include any description of the fan and flow rate used to aspirate the sensors.

    (b) There's also more sales pitchy statements such as 'patented sensor placement'.

    (c) There is also a lot of detail missing from the simulation setup. For example, what flow rates were used? What were the wind speeds? There are some large differences in Fig. 8 between the modeled system and the measured, which suggests that the modeled system is not capturing the dynamics of the environment and may not be providing useful guidance due to the missing factors.

    (d) Figure 7: I would argue that the air temperature within the duct would be much more useful information than the wall temperature, particularly if the sensors stand off from the wall. Also, the font in this figure legend is really small and hard to read.

    (e) Line 223: 'heated walls disrupt local airflow and introduce convective heating'. This statement is only true if the magnitude of the Richardson number is greater than order 1. The flow rate through the duct will affect this statement. What is the role of re-radiation in this environment?

    (f) Line 235: 'with the CS3D showing the larges absolute deviations'. What physical changes were made to the duct which would impact these deviations? Most of the changes described were to reduce airflow disturbance around the UAS.

    (g) Line 235: 'These observations confirm that solar loading is the dominant driver of sensor-to-sensor temperature discrepancies on both platforms'. I'm not entirely convinced this is the case given the issues mentioned above. Perhaps weaken this statement to 'suggest' instead of 'confirm'?

12. Line 263: Does not thrust, which is largely unknown, also need to be determined?

13. Line 268: The assumption that the mean wind changes slowly over time may be correct, but the drag will respond to rapid changes in the wind direction, not just the mean wind, so the assumption that $h = 0$ may be significant.

14. Lines 269-271 The connection between drag and wind is never expressed explicitly in this section. What relationship is used in this algorithm?

15. Line 304 'maintaining the best accuracy at the upper end of the wind speeds.' This doesn't indicate accuracy, instead should probably say something like 'best agreement with the references'.

16. Figure 11 What were the conditions under which these measurements were taken? It looks like this may be a developing convective boundary layer? In which case the fluctuations of the CS3D may actually indicate higher turbulence and not just noise as implied by the description (i.e. the actual instantaneous wind). The lidar results seem really smooth though, was any averaging applied to the lidar or is this an instantaneous profile?

17. Line 335: What is meant by the phrase 'notably tighter alignment'?

18. Line 346 What is an 'acceptable' error envelope? What defines this envelope?

19. Figure 14: The font is really small on these figures.

20. Figure 16 and associated text. The fact that the Coptersonde SWX produced readings in moderate rain/highly humid conditions does not imply that the readings were correct. Were there any reference measurements made (e.g. from a ground-based system) that can be used to confirm that the values in the profiles are actually correct?

21. Conclusions: There is more sales-pitchy language in the Conclusions. For example the statement 'setting it apart from conventional platforms' is made without support. There are other platforms which have been tested in severe weather (e.g. hurricane measurements have been made using UAS). Similarly, Line 458 mentions 'penetrating and characterizing extreme PBL conditions'. The weather conditions (high winds) which the aircraft was tested in are certainly challenging, but there are surely more 'extreme' weather conditions (particularly with regards to confounding factors like low temperature or heavy precipitation) which would would be better qualified as being extreme. Line 452 mentions 'Severe-weather flights' with regards to the tested conditions, but does a low level jet and moderate rain count as severe weather flights? There is no doubt that this aircraft possesses improved performance characteristics relative to its predecessors, but the paper would be better served by more reserved and precise language when describing its capabilities.